

# Predicting social media users' indirect aggression through pre-trained models

Zhenkun Zhou[1], Mengli Yu[2,3,4], Xingyu Peng[5] and Yuxin He[1]

[1] Department of Data Science, School of Statistics, Capital University of Economics and Business, Beijing, China
[2] School of Journalism and Communication, Nankai University, Tianjin, China
[3] Convergence Media Research Center, Nankai University, Tianjin, China
[4] Publishing Research Institute, Nankai University, Tianjin, Tianjin, China
[5] State Key Lab of Software Development Environment, Beihang University, Beijing, China

## ABSTRACT

Indirect aggression has become a prevalent phenomenon that erodes the social media environment. Due to the expense and the difficulty in determining objectively what constitutes indirect aggression, the traditional self-reporting questionnaire is hard to be employed in the current cyber area. In this study, we present a model for predicting indirect aggression online based on pre-trained models. Building on Weibo users' social media activities, we constructed basic, dynamic, and content features and classified indirect aggression into three subtypes: social exclusion, malicious humour, and guilt induction. We then built the prediction model by combining it with large-scale pre-trained models. The empirical evidence shows that this prediction model (ERNIE) outperforms the pre-trained models and predicts indirect aggression online much better than the models without extra pre-trained information. This study offers a practical model to predict users' indirect aggression. Furthermore, this work contributes to a better understanding of indirect aggression behaviors and can support social media platforms' organization and management.

## INTRODUCTION

Indirect aggression is traditionally regarded as "a kind of social manipulation: the aggressor manipulates others to attack the victim, or, by other means, makes use of the social structure in order to harm the target person, without being personally involved" (*Björkqvist, Lagerspetz & Kaukiainen, 1992*; *Brain, 1994*). In the current cyber era, the explosion of information technologies and social media has significantly changed the nature of social communication. Social media has become a global phenomenon in daily life. It allows to act more impulsively because they have the perception that their interactions are anonymous, thus enabling them to express their feelings freely in online social networks (*Bioglio & Pensa, 2022*). Consequently, social media has created a new channel for individuals to engage in indirect aggressive behaviors (*Mishna et al., 2018*; *Mladenović, Ošmjanski & Stanković, 2021*).

Indirect aggression online, a type of cyber social threat, refers to deliberate harmful behaviors directed towards others through online social networks (*Mladenović, Ošmjanski & Stanković, 2021*). This insidious phenomenon includes hostile actions like social

Corresponding author
Mengli Yu, mengliyu@nankai.edu.cn

exclusion, malicious humour, and guilt induction and poses significant challenges to the safety and well-being of individuals in digital spaces (*Mladenović, Ošmjanski & Stanković, 2021*; *Forrest, Eatough & Shevlin, 2005*). Previous studies have focused on aggressive behaviors on popular social network platforms like Facebook and Twitter and have demonstrated that user characteristics, such as age and gender, play a role in shaping the occurrences of cyber social threats (*Meter, Ehrenreich & Underwood, 2019*). For example, research has shown that males are more likely than females to engage in Facebook aggression, such as sending insulting messages and posting aggressive comments (*Bogolyubova et al., 2018*). Previous studies have primarily focused on specific groups such as adolescents and children. *Pabian, De Backer & Vandebosch (2015)* investigated the relationship between the Dark Triad personality traits and aggression on Facebook among adolescents. They found a significant relationship between the intensity exhibited during Facebook interactions and adolescents' aggressive behaviors.

Indirect aggression can lead to negative consequences, including the deterioration of relationships, substance use, rule-breaking behaviors, and even major criminal activity (*Mladenović, Ošmjanski & Stanković, 2021*; *Sadiq et al., 2021*). To address the negative impact of indirect aggression, researchers have been trying to identify and predict such harmful behaviors (*Sadiq et al., 2021*; *Sharif & Hoque, 2021*). Traditionally, indirect aggression has been measured through self-reporting, which has several limitations such as cost, subjectivity, and low flexibility. This weakens the quality of data and the validity of the conclusions drawn from it (*Zhou, Xu & Zhao, 2018*). The popularity of social media now provides a new avenue to explore users' personalities and psychological behaviors by providing vast amounts of data (*Yu & Zhou, 2022*; *Tovanich et al., 2021*; *Panicheva et al., 2022*). For instance, *Kosinski, Stillwell & Graepel (2013)* conducted a study in which they analyzed Facebook "Likes" to predict users' personality traits such as openness, conscientiousness, extraversion, agreeableness, and emotional stability. They found that patterns of "Likes" were significantly associated with these personality traits, showing the potential of the use of social media data to help understand individual traits. The analysis of online behavior has shown that social media content and interactions can provide valuable information about a user's psychological behaviors. *Schwartz et al. (2013)* investigated users' emotional expressions on Twitter and found that the content of the tweets could accurately predict a user's mood, including depression and anxiety levels. The study highlighted the relevance of using social media data to understand a user's emotional and psychological well-being. Additionally, research has explored the connection between social media language and mental health. *Coppersmith, Dredze & Harman (2014)* analyzed publicly available Twitter text data to predict mental health. The abundance of research on sentiment analysis and opinion mining also supports the notion that social media provides valuable data to explore psychological behaviors. The components of both a user's profile and posted images have been widely used to predict a user's depression and anxiety levels (*Guntuku et al., 2019*). This research showcases the ability of social media data to provide insights into users' psychological responses and behaviors in different contexts. The popularity of social media has indeed opened up new opportunities to explore users'

personalities and psychological behaviors through the vast amounts of data generated on these platforms.

Although there is relatively limited research specifically focused on indirect aggression, existing studies of direct aggression (*e.g.*, cyberbullying) offer relevant information that supports the idea of using social media data to detect such behaviors as shown in Appendix A. *Chavan & Shylaja (2015)* employed traditional feature extraction techniques like TF-IDF and N-grams to detect cyberbullying comments on social networks. *Al-Garadi et al. (2019)* explored feature selection and tested various machine learning algorithms to predict cyberbullying behaviors. However, it is important to note that indirect aggression differs significantly from direct aggression. Compared to predicting direct aggression, predicting indirect aggression, characterized by subtle tactics like implicit language, poses a greater challenge due to its covert nature (*Severance et al., 2013*). Indirect aggression, such as attempting to make others dislike somebody or to engage in gossiping, requires a higher level of social intelligence and social recognition compared to direct aggression (*Garandeau & Cillessen, 2006*). Furthermore, subtypes of indirect aggression, like social exclusion and malicious humour, involve different levels of social manipulation and rational-appearing aggression (*Forrest, Eatough & Shevlin, 2005*). It is crucial to distinguish between subtypes of indirect aggression online using advanced techniques. Conventional machine learning techniques are often limited when processing natural data in its raw form, requiring careful engineering and domain expertise (*LeCun, Bengio & Hinton, 2015*). With the current advancements in deep neural networks, large-scale pre-trained models have made significant strides in addressing these challenges (*Devlin et al., 2018*). Therefore, this study aims to propose a new method that combines the pre-trained model to predict users' indirect aggression online.

Indirect aggression online affects social media platforms' operations and the overall digital interactive environment. The vast amount of data available on social media, coupled with the advancements in computational social science, has opened new avenues for understanding individuals' personalities and behaviors. This study proposes a novel methodology that leverages large-scale pre-trained models' capabilities to handle complex and nuanced behaviors that traditional machine learning techniques struggle with. By incorporating basic, dynamic, and content features, our method provides a comprehensive analysis of indirect aggression behaviors. This model outperforms traditional machine learning approaches by effectively capturing the intricate patterns and subtleties within social media interactions. The significance of our proposed model lies in its ability to accurately detect covert forms of aggression that are often subtle and implicit. Moreover, our study provides clear guidelines for using pre-trained models like ERNIE to predict indirect aggression, offering valuable insights for researchers and practitioners in natural language processing and behavior analysis. Through rigorous testing and validation, we demonstrated the robustness and reliability of our methodology, which can be adapted and applied to various predictive tasks beyond aggression and facilitate the optimization of social media platforms. This study contributes to the growing field of computational social science by providing an innovative approach to understanding and mitigating negative behaviors in online environments, and fosters a safer and more harmonious digital space.

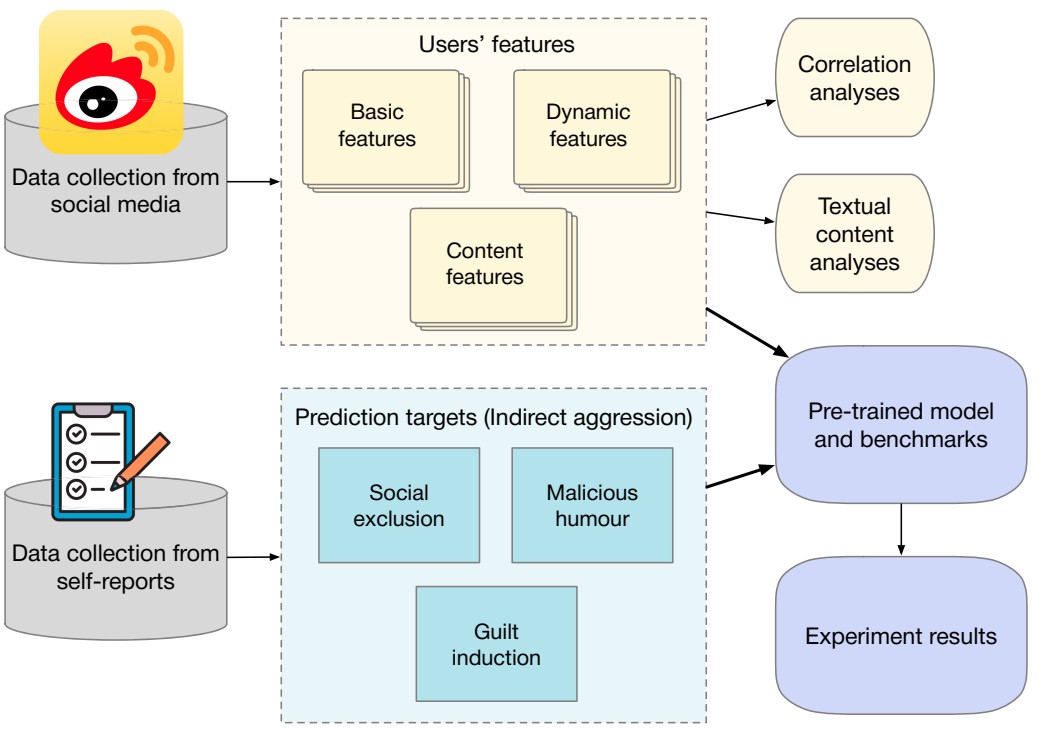

**Figure 1 Experiment steps.** Applications, media, social icon by tulpahn. CC BY-NC 3.0, https://www.
iconfinder.com/icons/4102588/applications_media_social_weibo_icon. Survey, rating, feedback icon by
Cyber Olympus Indonesia. CC BY 4.0, https://www.iconfinder.com/icons/10186533/survey_rating_
feedback_review_checklist_rate_questionnaire_icon.   

## DATASET

Portions of this text were previously published as part of a preprint (*Zhou et al., 2023*).

### Participants and data collection

The steps in this study are illustrated in Fig. 1. This study was reviewed and approved by
the Human Research Ethics Committee of the School of Economics and Management at
Beihang University. The study used the online surveys for data collection. Students from
the University of Beijing in China were invited to participate in this survey. Informed
consent was obtained from all participants. The consent process was conducted in written
form, ensuring participants were fully informed about the study's purpose, procedures,
and their rights. A total of 456 valid questionnaires were collected. All participants
provided their Weibo IDs, enabling us to collect their Weibo activities and profile data. To
guarantee the quality of the user data, we only select active users with more than 20 Weibo
tweets. The selected users must have posted since 2020 and must have been obviously
active for at least 2 months on Weibo. Three hundred and twenty active accounts are
selected for this research on indirect aggression in social media, including 74 males and
246 females.

We employ the Indirect Aggression Scale Aggressor model (*Mladenović, Ošmjanski &
Stanković, 2021*; *Forrest, Eatough & Shevlin, 2005*) to measure participants' indirect
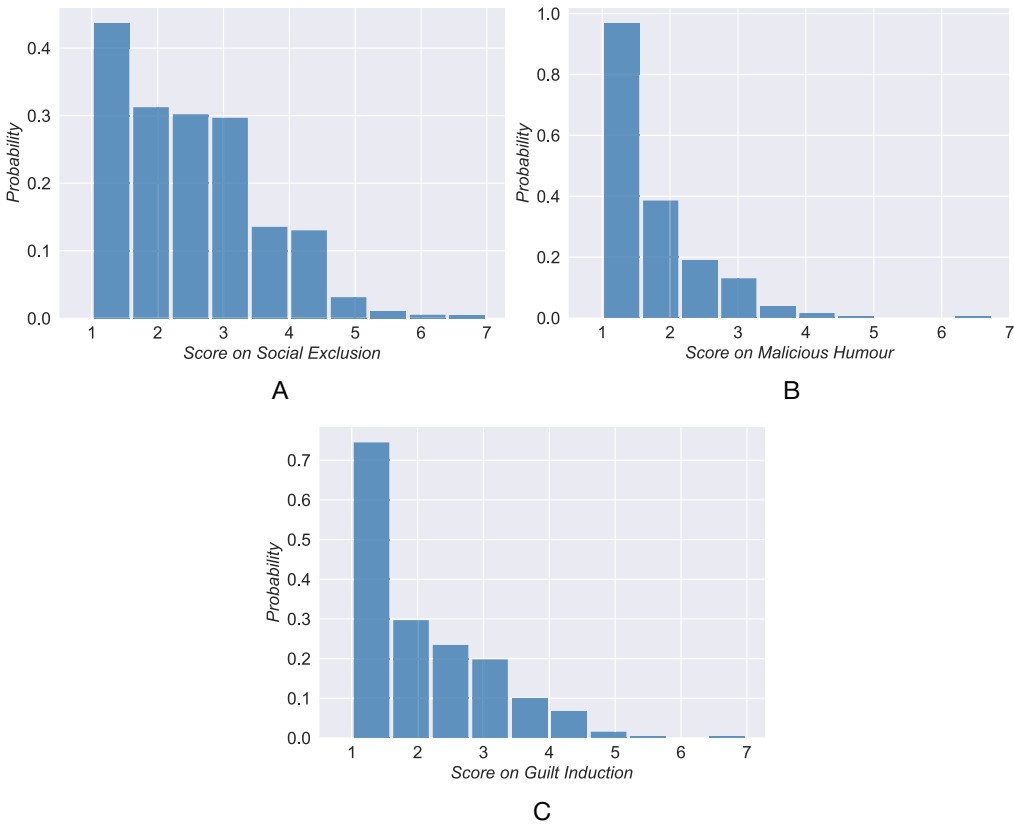

**Figure 2 The distribution of self-reports scores of active users.**

aggressive behaviors, including three types of social exclusion (10 items), malicious humour (nine items), and guilt induction (six items) strategies. Specifically,

- social exclusion refers to behaviors that work by socially excluding the victim, such as withholding information, leaving out of activities, and turning the community against someone;
- malicious humour refers to behaviors in which humour is used to harm the victim, such as the use of sarcasm as an insult, intentional embarrassment, and practical joke playing;
- guilt induction refers to behaviors whereby guilt is intentionally induced, such as the use of emotional blackmail, undue pressure, and coercion.

Participants are first told to "Please answer the questions below according to how you interact with your contacts on Weibo." The answers are measured using 7-point Likert scales, from "1 = Never behave this way" to "7 = Always behave this way".

Three subscales of Alpha coefficients range from 0.76 to 0.92. The social exclusion ($\mu = 2.44$, $\sigma = 1.11$), malicious humour ($\mu = 1.69$, $\sigma = 0.77$), and guilt induction ($\mu = 2.03$, $\sigma = 1.02$) are obtained by computing the average of each relevant item ($\mu$ as Mean, $\sigma$ as SD).

**Table 1 The cluster centroids (samples) in categories.**

|  | Low group | Neural group | High group |
| --- | --- | --- | --- |
| Social exclusion | 1.21 (103) | 2.60 (156) | 4.14 (61) |
| Malicious humour | 1.16 (178) | 2.05 (106) | 3.34 (36) |
| Guilt induction | 1.17 (152) | 2.39 (117) | 3.81 (51) |

## Subjects categoring

Inspired by previous studies (*Zhou, Xu & Zhao, 2018*; *Yu & Zhou, 2022*), it is reasonable to divide the subjects into three categories (high, neutral, and low) according to the psychological prediction goals. The distribution of scores unexpectedly follows a long-tail distribution instead of a Gaussian distribution (Fig. 2). This implies that most individuals have low scores and few have high scores. The clustering algorithm K-means is applied to classify the individuals' scores into three categories. The subject is grouped into the closest category according to the distance from the cluster centroid. Table 1 shows the cluster centroid and the number of users in each category. By classifying 320 active users into three categories, we obtain a training set used next to analyze and establish the prediction models.

## METHOD

Portions of this text were previously published as part of a preprint (*Zhou et al., 2023*).

### Users' features

Using the three prediction targets, namely social exclusion, malicious humour, and guilt induction, we extract and analyze three groups of features in order to determine a user's characteristics. These features serve as inputs for the prediction model. The first and second group of features, respectively, contain a user's basic and dynamic features. The third group primarily depicts the text in a user's tweets. We calculate the three groups of features below.

Basic features $X_{b1} \sim X_{b13}$ reflect the user's demographics and static numeric information on social media. The demographics $X_{b1} \sim X_{b4}$ include gender (male or female), educational background, occupation and hobbies. Additionally, the static numeric information includes $X_{b5} \sim X_{b13}$:

- $log(DAYS + 1)$, where $DAYS$ represents the number of days from the user's registration to the survey;
- $log(NT + 1)$, where $NT$ represents the total number of the user' tweets;
- $log(NT/(DAYS + 1))$;
- $log(NF + 1)$, where $NF$ represents the number of the user's followers;
- $log(NFE + 1)$, where $NFE$ represents the number of the user's followees;
- $log((NFE + 1)/(NF + 1))$;
- $NT/(NF + 1)$;

- $NT/(NFE + 1)$;
- the length of the user's self-description text.

Dynamic features $X_{d1} \sim X_{d125}$ are designed to reflect the intricate patterns of social interactions on Weibo both daily and weekly. Social interaction refers to posting, mentioning, and retweeting, which are key behaviors related to psychological traits in previous research. First, we calculate hourly and daily feature vectors based on the occurrences of interactions at each hour of the day and each day of the week. Here, hourly dynamic feature vector contains 24-dimensional hour-level feature and daily dynamic feature vector contains 7-dimensional day-level feature. These vectors are integral components of the dynamic features.

Second, we extract statistical measures from the hour-level and day-level features. For example, the following five features of "posting behavior" are extracted from the hour-level feature vector:

- the average hourly posting counts;
- the maximum of hourly posting counts;
- the hour with the most posts;
- the hour with the least posts;
- the variance of posting counts at different hours.

The proportions of tweets with mentions and retweets are also considered as features that reflect a user's intensity of interaction.

User-generated content provides rich information about a user's behaviors, which can also reflect their psychological traits. Previous studies on personality traits show that a user's language styles on social media can effectively predict psychological traits (*De Choudhury et al., 2013*; *Panicheva et al., 2022*). In our study, we first utilize 300-dimensional pre-trained word embeddings $X_{w1} \sim X_{w300}$ to represent each word in Weibo. The word embeddings are taken from the pre-trained Chinese Word Vectors using Positive Pointwise Mutual Information (PPMI) (*Li et al., 2018*). We average the word embeddings to create the content vector at the Weibo level and subsequently average the Weibo-level vectors to obtain 300-dimensional content vectors at the user level. It should be noted that the text in a user's descriptions is also considered part of the content features.

The extraction results in a 13-dimensional basic feature ($X_{b1} \sim X_{b13}$), 125-dimensional dynamic feature ($X_{d1} \sim X_{d125}$), and 300-dimensional content feature ($X_{w1} \sim X_{w300}$). We extract 512-dimensional content vector representations ($X_{p1} \sim X_{p512}$) from the text content using pre-trained models, which are detailed in the Models section.

## Correlation analyses between features and indirect aggression

Based on the aforementioned user's features, we first compute and examine the Pearson correlation coefficients between features of different categories. Applying a coefficient threshold of 0.1, we find that an individual's scores of indirect aggression from the three categories is related with both the basic and dynamic features ($p$-value $< 0.05$).

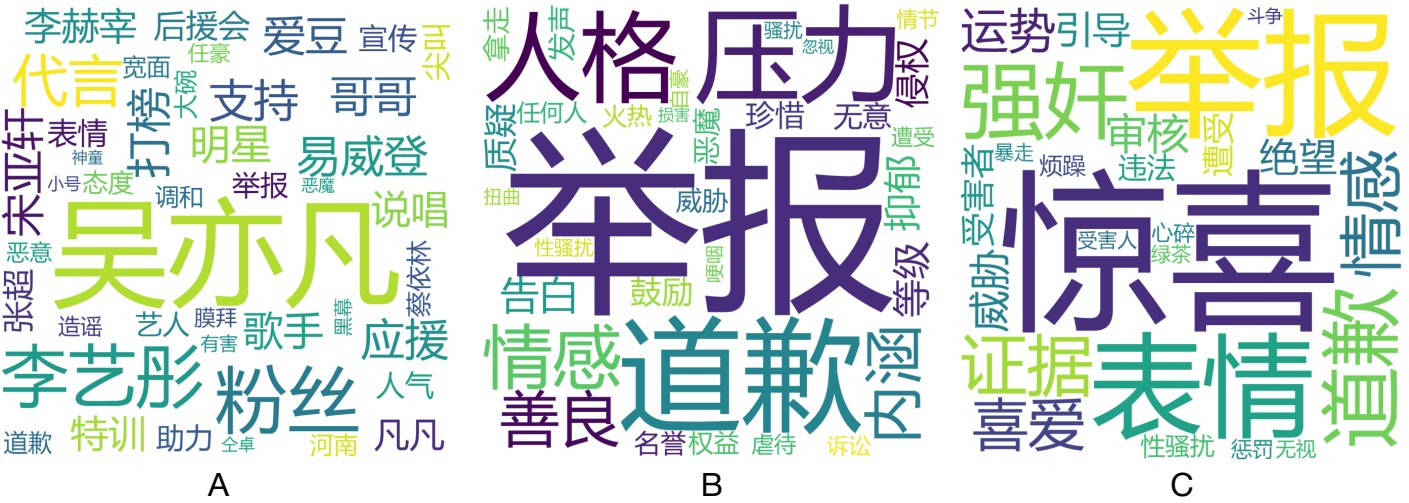

**Figure 3** (A–C) Word cloud for social exclusion, malicious humour and guilt induction.

Gender, as a demographic characteristic, emerges as one of the most significant features of malicious humour (Coef. = 0.240) and guilt induction (Coef. = 0.147). In terms of basic interactions, social exclusion shows a positive correlation with the number of followers (Coef. = 0.125), suggesting that users experiencing high social exclusion have more followers. When looking at dynamic features, social exclusion and malicious humour are positively correlated with social media activities (Coef. = 0.958 and Coef. = 0.119, respectively), such as tweeting frequency at different times. We observe that the three categories significantly and positively correlate with individual mentions and retweeting activity.

## Textual content analyses

From the perspective of the text content, key words in three subcategories are computed and analyzed. We choose the significant word results from the word frequency in the high group and the low group (shown in Table 1) using analysis of variance. The word cloud in Fig. 3 depicts the key words and the size of the words represent the significance (1/$p$-value). It also shows the different emotional levels and aggressive behaviors.

Instances of social exclusion are direct and aggressive, requiring lower levels of social manipulation. The words such as "accusation" (举报), "threaten" (威胁), "rape" (强奸), and "victim" (受害者) are used to express exclusion. The expression of malicious humour is rational-appearing aggression with the use of euphemistic and gentle words and thus requires a higher-level social manipulation. The names of tarnished celebrities, such as "Yifan Wu" (吴亦凡), "Yaxuan Song" (宋亚轩), and "Yitong Li" (李艺彤) are used to express maliciousness due to their controversial behaviors or evil doings. Finally, guilt inducting expressions apply psychological pressure, prompting individuals to reflect on their emotions and behaviors. This level of social manipulation falls between that of social exclusion and malicious humour. The words like "apologize" (道歉), "pressure" (压力), and "connotation" (内涵) express guilt.

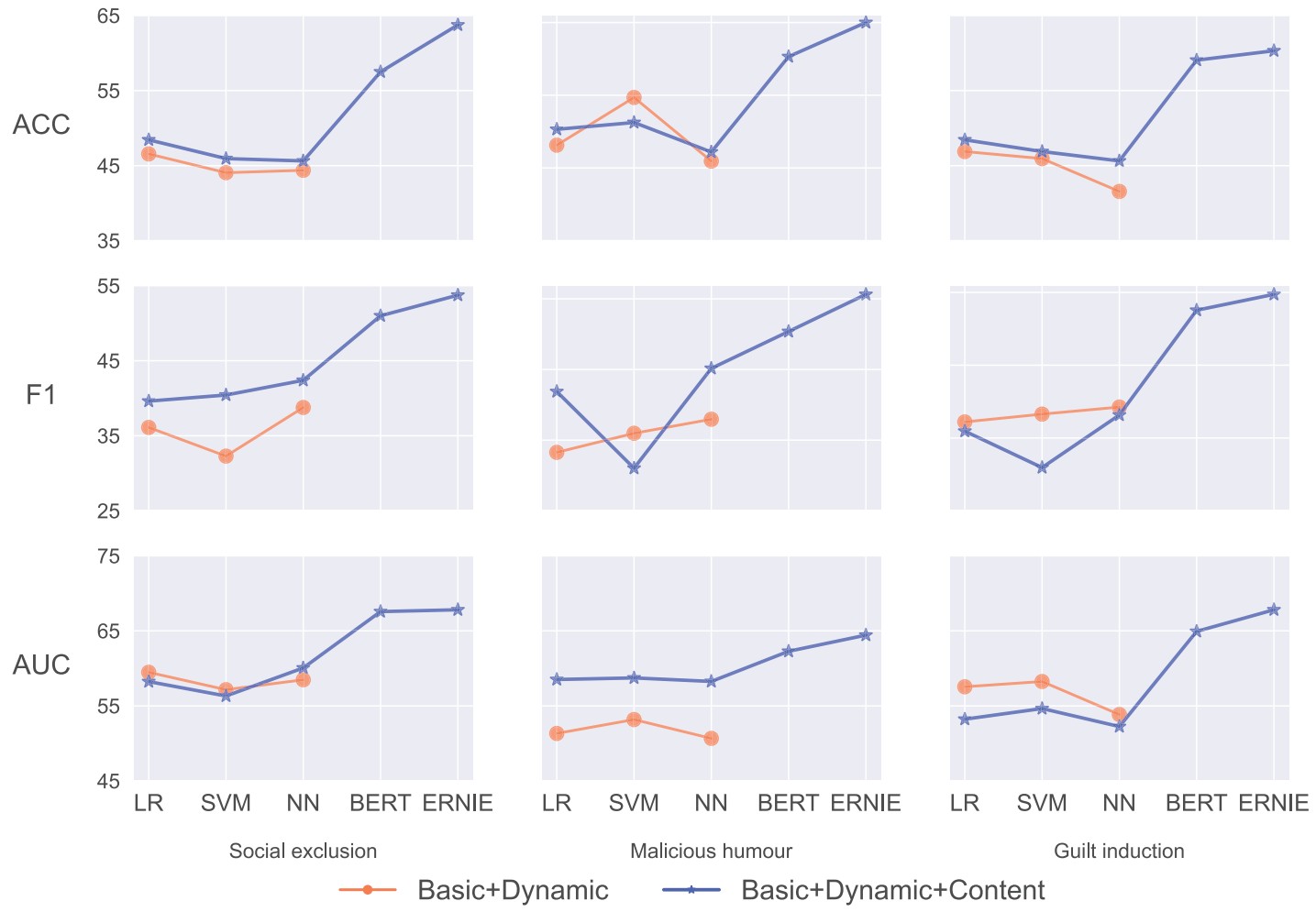

**Figure 4** The model architecture of ERNIE (cited from *Sun et al. (2021)*).

## Models

Recently, we have witnessed the fast growth of large-scale pre-trained models in the natural language processing field. They are often pre-trained with an extremely large amount of data and have demonstrated good performance across various prediction tasks (*Vaswani et al., 2017*; *Han, Zhang & Ding, 2021*; *Min et al., 2023*). The Transformer architecture has the potential to represent text content, which can substantially improve the research on psychological prediction. Bidirectional Encoder Representations from Transformers (BERT) (*Devlin et al., 2018*), as a pre-trained model, was used to generate the text represented for each social media user. We first utilized the pre-trained BERT based on the Chinese *corpus* to predict the users' indirect aggression in our study.

However, the existing pre-trained language models rarely integrate knowledge graphs, which offer rich structured knowledge and entities to enhance understanding of natural language. Consequently, we use ERNIE, a sophisticated modification of the BERT architecture, to incorporate linguistic and world knowledge (*Sun et al., 2021*). Figure 4

shows the architecture of ERNIE. The current version, ERNIE 3.0, introduces a framework for large-scale pre-trained knowledge-enhanced models with 175 billion parameters. Empirical results demonstrate that the ERNIE 3.0 model outperforms the state-of-the-art models on multiple Chinese NLP tasks.

As shown in Fig. 4 (right), the knowledgeable encoder (K-Encoder) is a key module in this model. It is designed to effectively integrate heterogeneous information from the content of the text and from the knowledge graphs. The K-Encoder integrates additional token-oriented knowledge into content features at the attention layer. It consists of stacked aggregators where the input word embeddings $\{w_1^i, \ldots, w_n^i\}$ and entity embeddings $\{e_1^i, \ldots, e_n^i\}$ from the previous aggregator are fed into multi-head self-attentions at the $i$-th aggregator. The operation of $i$-th aggregator operation is denoted as follows:

$$\{w_1^{i+1}, \ldots, w_n^{i+1}\}, \{e_1^{i+1}, \ldots, e_n^{i+1}\} = \text{Aggregator}(\{w_1^i, \ldots, w_n^i\}, \{e_1^i, \ldots, e_n^i\}). \tag{1}$$

Specifically, the $i$-th aggregator adopts an information fusion layer for combining the token and entity sequence, and computes the output embedding for each token and entity. For a token $w_j$ and its aligned entity $e_k$, the information fusion process is as follows,

$$\begin{aligned} h_j &= \sigma(\widetilde{W}_t^i \widetilde{w}_j^i + \widetilde{W}_e^i \widetilde{e}_k^i + \widetilde{b}^i), \\ w_j^i &= \sigma(W_t^i h_j + b_t^i), \\ e_k^i &= \sigma(W_e^i h_j + b_e^i) \end{aligned} \tag{2}$$

Here, $h_j$ is the hidden state that integrates the information from both the token and the entity. $\widetilde{w}_j^i$ and $\widetilde{e}_k^i$ are output from multi-head self-attentions with input of $w_j^i$ and $e_k^i$.

The output computed by the last aggregator serves as the final output embeddings of the K-Encoder. This allows us to input the heterogeneous information of words and entities from user-generated content into the content vectors. The information fusion can enhance the prediction of users' indirect aggression.

Moreover, an increasing number of promising pre-trained models based on Transformer architecture have emerged including BART (*Lewis et al., 2019*), RoBERTa (*Liu et al., 2019*), and ELECTRA (*Clark et al., 2020*). Researchers evaluate the design decisions of these pre-trained models. The findings indicate that performance can be improved by training the model longer, with bigger batches, and by dynamically changing the masking patter applied to the training data. This leads to the proposal of RoBERTa (*Liu et al., 2019*). In addition, ELECTRA outperforms MLM-based models (masked language modeling) such as BERT and RoBERTa when considering the same model size and data. This is attributed to its compute-efficient and parameter-efficient ability to distinguish real data from challenging negative samples (*Clark et al., 2020*). We use the competitive models RoBERTa and ELECTRA for comparison with BERT and ERNIE in this study.

In this study, we compute the average word embeddings from large pre-trained models that generate 512-dimensional content embeddings to represent the users' behavior. In the output layer, we incorporate both Basic + Dynamic features in order to fine-tune the

model. We aim to develop the ultimate prediction model. The experiments are performed with PyTorch (v1.12) and Transformers (v4.7.0) on a Linux server equipped with two GPUs (NVIDIA V100) and two CPUs (Intel Xeon Silver 4216). The anonymous dataset and code have been released publicly. The pre-trained language models in this study are obtained from HuggingFace.

# EXPERIMENT AND RESULTS

## Experimental design

In this section, we employ various models to predict and assess the predictive power regarding three specific forms of indirect aggression among social media users: social exclusion, malicious humour, and guilt induction.

The experiment is structured in two phases: the first involves pre-trained and benchmark models; the second evaluates advanced pre-trained models. Initially, we compare benchmark models such as LR, SVM, and MLP with pre-trained models like BERT and ERNIE, demonstrating that the latter significantly outperform traditional machine learning approaches. This phase evaluates the predictive power of user-generated content using two feature sets: Basic + Dynamic and Basic + Dynamic + Content.

Subsequently, we include additional advanced pre-trained models, RoBERTa and ELECTRA, to identify the most effective model for predicting users' indirect aggression. This comprehensive analysis not only underscores the superior performance of advanced models but also enhances our understanding of their applicability in real-world scenarios. Performance evaluation for each model is rigorously conducted using the dataset split into 70% training and 30% testing segments, facilitating robust validation.

## Evaluation metrics

The prediction performance of the aforementioned models is verified by Accuracy ($ACC$), F1-score ($F1$), and $AUC$ (Area Under the Receiver Operating Characteristic Curve). We build 3-category classifiers for users' indirect aggression and then obtain true positive ($TP$), true negative ($TN$), false positive ($FP$), and false negative ($FN$) for the category $i = -1, 0, 1$. $N_{samples}$ is the total number of samples. The expressions for $ACC$ and $F1$ are below.

$$ACC = \frac{\sum_{i=-1,0,1} TP_i}{N_{samples}}. \tag{3}$$

Calculations for the precision and recall for each label $i$ are below:

$$Precision_i = \frac{TP_i}{TP_i + FP_i}, \tag{4}$$

$$Recall_i = \frac{TP_i}{TP_i + FN_i}, \tag{5}$$

$$F1 = \frac{1}{3} \sum_{i=-1,0,1} \frac{2(Pr_i \times Re_i)}{Pr_i + Re_i}. \tag{6}$$

**Table 2 The performance (%) for three category models for predicting indirect aggression online: LR, SVM, MLP, BERT, and ERNIE (bold for the best performance).**

| Prediction target | Features | Model | ACC | F1 | AUC |
|---|---|---|---|---|---|
| Social exclusion | Basic + Dynamic | LR | 46.56 | 36.11 | 59.48 |
| | | SVM | 44.06 | 32.29 | 57.16 |
| | | MLP | 44.38 | 38.77 | 58.48 |
| | Basic + Dynamic + Content | LR | 48.44 | 39.62 | 58.24 |
| | | SVM | 45.94 | 40.43 | 56.32 |
| | | MLP | 45.63 | 42.39 | 60.06 |
| | | BERT | 57.50 | 50.99 | 67.57 |
| | | **ERNIE** | **63.75** | **53.74** | **67.81** |
| Malicious humour | Basic + Dynamic | LR | 48.13 | 33.27 | 51.33 |
| | | SVM | 54.69 | 35.97 | 53.18 |
| | | MLP | 45.94 | 37.96 | 50.67 |
| | Basic + Dynamic + Content | LR | 50.31 | 41.88 | 58.52 |
| | | SVM | 51.25 | 31.03 | 58.73 |
| | | MLP | 47.19 | 45.16 | 58.26 |
| | | BERT | 60.31 | 50.38 | 62.28 |
| | | **ERNIE** | **65.00** | **55.63** | **64.43** |
| Guilt induction | Basic + Dynamic | LR | 46.88 | 37.21 | 57.55 |
| | | SVM | 45.94 | 38.28 | 58.25 |
| | | MLP | 41.56 | 39.25 | 53.85 |
| | Basic + Dynamic + Content | LR | 48.44 | 35.94 | 53.23 |
| | | SVM | 46.88 | 30.94 | 54.65 |
| | | MLP | 45.63 | 38.16 | 52.25 |
| | | BERT | 59.06 | 52.55 | 64.93 |
| | | **ERNIE** | **60.31** | **54.73** | **67.81** |

The *AUC* of each category can be measured against the rest of the categories, namely the one-*vs*-rest *AUC* (*Li et al., 2020*). *F*1 and *AUC* serve as proxies for model validation even though the testing dataset is not unbalanced.

## Results of pre-trained models and benchmark models

Table 2 details the performance of various models in predicting users' indirect aggression within the three categories: social exclusion, malicious humour, and guilt induction. The models include LR, SVM, and MLP as the benchmarks, and BERT and ERNIE represent the pre-trained models. The models are evaluated based on two groups of features: Basic + Dynamic and Basic + Dynamic + Content.

The BERT and ERNIE models' performance with Basic + Dynamic + Content features is outstanding in predicting social exclusion. They achieve *ACC* = 57.50%, *F*1 = 50.99%, *AUC* = 67.57% and *ACC* = 63.75%, *F*1 = 48.48%, *AUC* = 61.88% respectively. These two pre-trained models, especially ERNIE, significantly outperform MLP and other benchmark models.

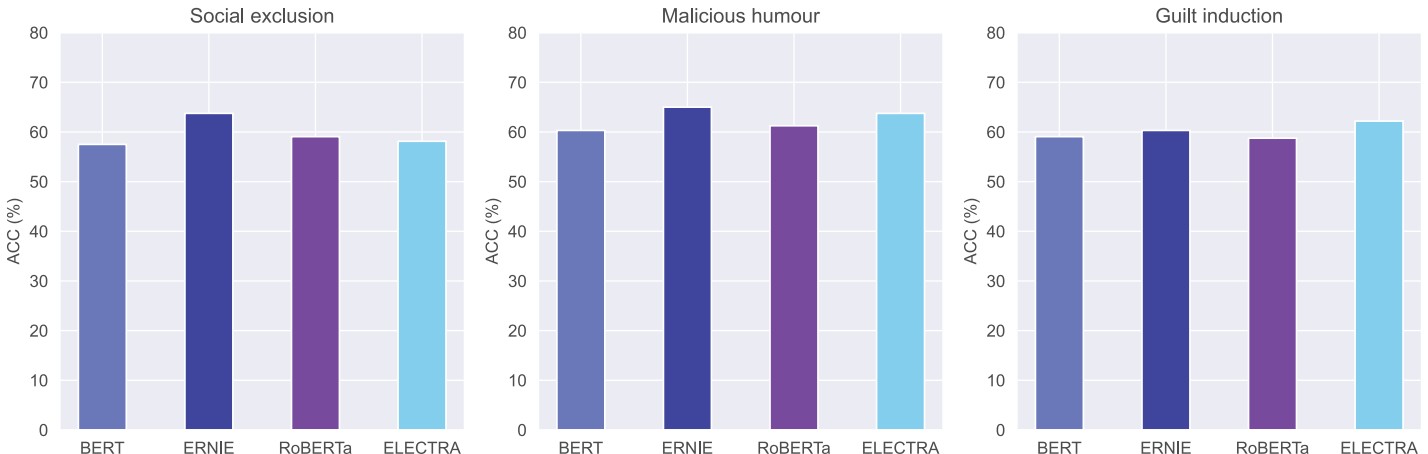

**Figure 5** **Model performance comparison for social exclusion, malicious humour and guilt induction.**

While using only Basic + Dynamic features, the SVM model leads with an accuracy of 54.69% and *AUC* of 53.18% for predicting malicious humour. With the addition of content features, there is an improvement in the LR's performance, which shows an accuracy of 50.31%. MLP achieves the highest *F*1 score at 45.16%. The pre-trained models surpass these results, with ERNIE leading with an *ACC* of 65.00%, an *F*1 of 55.63% and an *AUC* of 64.43%.

The pre-trained models demonstrate superior performance in predicting guilt induction, with ERNIE achieving the highest accuracy of 60.31%, an *F*1 score of 54.73%, and an *AUC* of 67.71%. The Basic + Dynamic features yield the highest accuracy with LR at 46.88% and the highest *AUC* score of the SVM at 58.25%. When incorporating content features, the LR's accuracy improves to 48.44%, but the *F*1 scores drop across all models.

Figure 5 shows that BERT and ERNIE with Basic + Dynamic + Content features have the most noticeable performance improvement compared with the other benchmarks, including prediction in the three categories of social exclusion, malicious humour, and guilt induction. We obtain these results consistently. Pre-trained models perform consistently highest in our study.

We find ERNIE is significantly better than BERT because of its richly structured knowledge. ERNIE substantially raises the prediction performance, improving 15.31%, 13.31%, and 11.87% of the *ACC* compared with the best baseline model. The results of the *F*1 score and the *AUC* support this conclusion as well. ERNIE consistently shows the best performance across all metrics and categories, indicating its effectiveness in understanding and predicting indirect aggression online.

When adding the content features to the traditional machine learning models, there is limited improvement, as shown by the blue and orange lines in Fig. 5. The *ACC*, *F*1 score, and the *AUC* of the MLP with Basic + Dynamic + Content features (45.63%, 42.39%, 60.06%) are slightly better (+1.25%, +2.77%, and +1.58%) than those of the MLP with

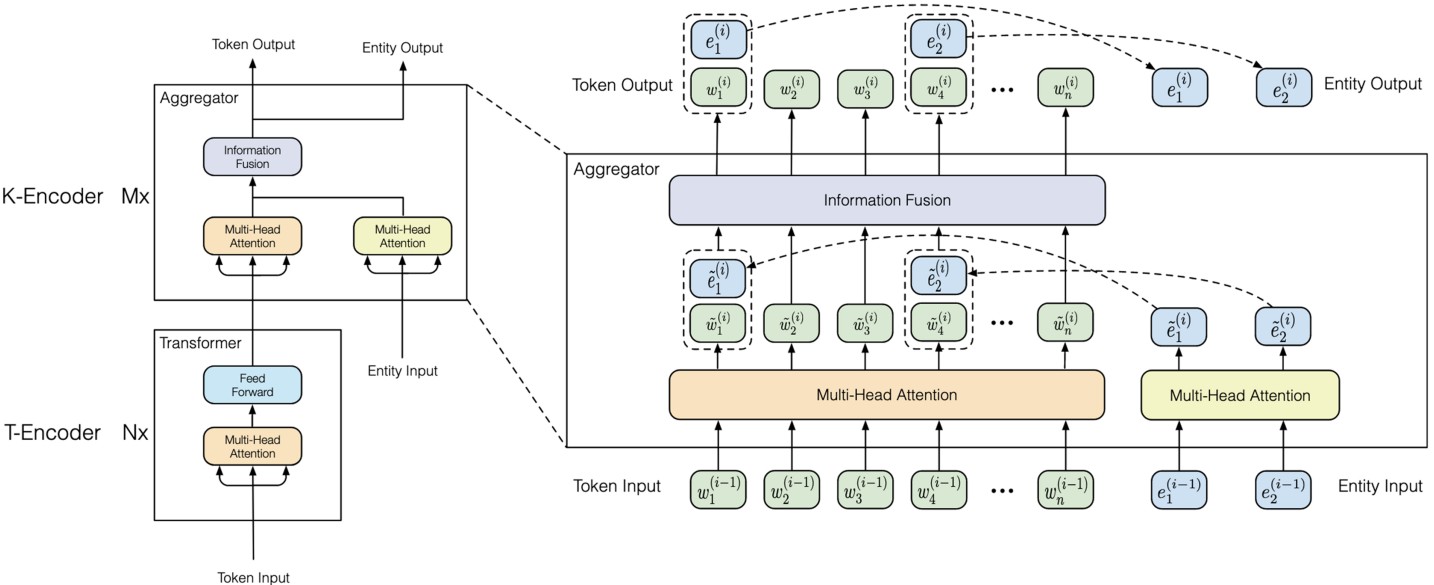

**Figure 6** **The accuracy (%) for pre-trained models for indirect aggression online: BERT, ERNIE, RoBERTa, and ELECTRA.**

Basic + Dynamic features. This demonstrates that the text content cannot be adequately mined by the simple and traditional word embeddings normally used for predicting psychological traits.

Additionally, we evaluate the time efficiency of the prediction models. Our optimal model leverages the pre-trained ERNIE model, significantly reducing the need for extensive training from scratch. This combination of pre-trained models and optimized feature extraction enhances both prediction accuracy and efficiency. Specifically, ERNIE completes training in 68.12 s and achieves an inference rate of 8.43 users per second. In comparison, the MLP model completes training in 35.33 s with an inference rate of 6.34 users per second. This demonstrates that in our study, ERNIE requires only slightly more training time compared to the traditional machine learning solution. Considering application scenarios of psychology, this time efficiency is acceptable.

### Results of advanced pre-trained models
We argue that leveraging advanced pre-trained models can increase the potential of automating textual coding and potentially improve the ability to predict indirect aggression. The fact that the model is trained on a small batch of samples, a concept commonly referred to as few-shot learning, should not be overlooked. In addition to the BERT and ERNIE models, we construct and evaluate two pre-trained models: RoBERTa and ELECTRA.

Figure 6 presents a clear comparison of the accuracy of the four models across the three prediction targets. ERNIE demonstrates the highest accuracy in predicting users' social exclusion and malicious humour behaviors, achieving an accuracy of 63.75% and 65.00%.

**Table 3 The performance (%) of pre-trained models in predicting online indirect aggression: BERT, ERNIE, RoBERTa, and ELECTRA (bold for the best performance).**

| Prediction target | Model | ACC | F1 | AUC |
|---|---|---|---|---|
| Social exclusion | BERT | 57.50 | 50.99 | 67.57 |
| | ERNIE | **63.75** | **53.74** | 67.81 |
| | RoBERTa | 59.06 | 53.47 | **69.07** |
| | ELECTRA | 58.13 | 49.63 | 67.81 |
| Malicious humour | BERT | 60.31 | 50.38 | 62.28 |
| | ERNIE | **65.00** | 55.63 | 64.43 |
| | RoBERTa | 61.25 | **57.42** | 63.19 |
| | ELECTRA | 63.75 | 51.14 | **64.69** |
| Guilt induction | BERT | 59.06 | 52.55 | 64.93 |
| | ERNIE | 60.31 | **54.73** | **67.81** |
| | RoBERTa | 58.75 | 51.19 | 65.58 |
| | ELECTRA | **62.19** | 51.22 | 65.75 |

Although ELECTRA achieves a slightly higher accuracy rate (62.19%) than ERNIE (the second-best 60.31%) for predicting guilt induction, ERNIE achieves the highest *F*1 score (54.73%) and *AUC* (67.81%).

Table 3 shows the comprehensive performance of four pre-trained models (BERT, ERNIE, RoBERTa, and ELECTRA) across three categories of indirect aggression online. ERNIE consistently demonstrates superior performance scores across all categories of indirect aggression online. Informative entities, such as entertainment stars and controversial figures, are often the targets of indirect aggression. ERNIE's knowledge graphs contain rich information about entities and the relationships between them. The results imply that knowledge graphs play a pivotal role in predicting indirect aggression, thereby contributing to the enhancement of the model's performance.

## DISCUSSION AND CONCLUSION

In this study, we accurately predict human indirect aggression online by harnessing the power of social media users' activities through the implementation of the pre-trained model. Our approach includes the categorization of three distinct behaviors associated with indirect aggression: social exclusion, malicious humour, and guilt induction. The implications of our research hold significant theoretical and practical value.

### Theoretical contribution

This study makes a notable theoretical contribution by expanding the understanding of indirect aggression. By going beyond the conventional reliance on self-reporting methodologies, we addressed the need for innovative approaches to predict indirect aggression using pre-trained models. This pioneering step opens up new possibilities for examining and comprehending the complexities of indirect aggressive behaviors in online social media contexts. By harnessing the power of computational techniques, we provide a

fresh perspective on indirect aggression, shedding light on how it manifests itself and the nuances involved that were previously challenging to discern. Our research offers clear guidelines for using the pre-trained model to predict indirect aggression. Incorporating basic, dynamic, and content features, we provide insights into the roles these features play in predicting different types of indirect aggression. These guidelines not only contribute to the specific task of predicting indirect aggression but also have a broader application in the field of natural language processing and behavior analysis. Researchers and practitioners can benefit from this structured approach to harnessing ERNIE's capabilities in various predictive tasks beyond aggression.

A significant theoretical finding of our study is the outperformance of pre-trained models compared to traditional machine learning models that lack external pre-trained information. Demonstrating the superiority of pre-trained models like ERNIE in predicting social media users' behaviors provides compelling evidence of the effectiveness of using these pre-trained techniques to capture the intricate patterns and subtleties found within social media interactions. We present three reasons why ERNIE is suitable for our prediction research. First, ERNIE often surpasses BERT in few-shot learning and fine-tuning with limited data (*Sun et al., 2019*; *Zhang et al., 2019*). Second, informative entities, such as entertainment stars and controversial figures, are often targets of indirect aggression. ERNIE's knowledge graphs contain rich information about entities and their relationships, improving predictions of users' indirect aggression. Third, as shown in Fig. 4, the K-Encoder integrates token-oriented knowledge into textual information from the underlying layer, representing heterogeneous information of tokens and entities in a unified feature space. This integration can enhance the prediction of indirect aggression. This result extends the theoretical boundaries of prediction models to other psychological contexts, particularly when dealing with limited data samples, and highlights the potential of pre-trained models to advance advancing behavioral research in the digital age.

Additionally, the novelty of our dataset lies in its extensive collection and annotation of indirect aggression instances on social media platforms, which is a relatively unexplored area. The innovative aspects of our method include the integration of multiple feature types and the application of a sophisticated pre-trained model, which together significantly enhance prediction accuracy and reliability. These contributions not only fill a gap in the current literature but also set a foundation for future research to build upon.

## Practical implications

From a pragmatic standpoint, our research offers a workable system for predicting indirect aggression online (shown in Fig. 7). By moving beyond the limitations of conventional self-reporting questionnaires, our pre-trained model, based on ERNIE 3.0, provides a more objective and data-driven means to detect and understand indirect aggressive behaviors online.

Social media platforms can leverage our findings and methodology to implement proactive measures for identifying and addressing instances of indirect aggression. This can enhance their capacity to promote a safe and respectful online environment. Additionally, this study extends to social media platforms' organizational strategies. By

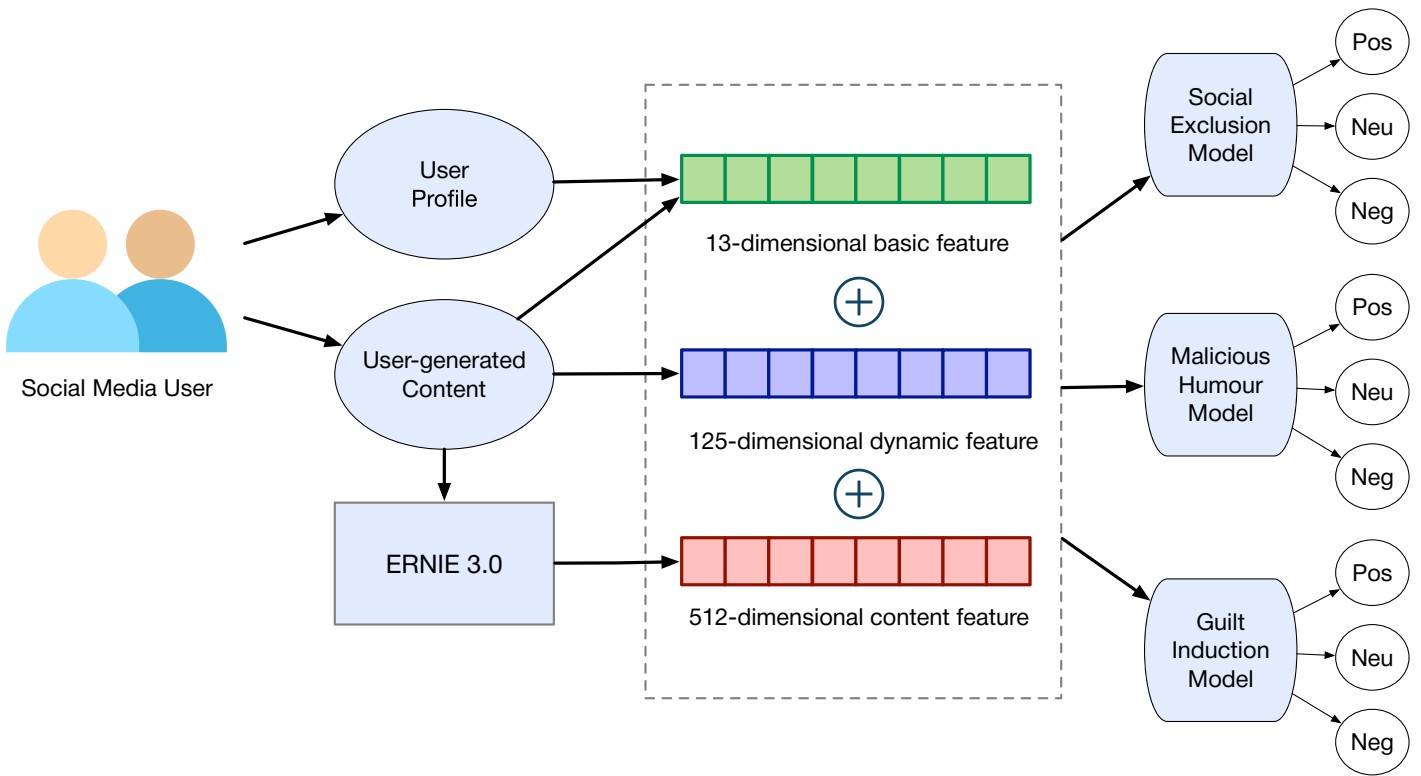

**Figure 7 The system architecture.** Add, contact, user icon by Flatart. CC BY 3.0, https://www.iconfinder.com/icons/5172568/add_contact_user_icon.

monitoring and recognizing users' indirect aggression through our predictive model, platforms can intervene early to mitigate potential harmful behaviors. This proactive approach empowers platforms to take appropriate actions, such as providing supportive interventions, issuing warnings, or offering educational resources to users engaging in indirect aggression.

As a result, fostering a more inclusive and respectful user experience becomes feasible, with the potential to reduce instances of cyberbullying and negative interactions. In addition, the optimization of platforms' organizational strategies, based on our predictive model, can lead to a more positive user experience and improved well-being for their user base. By curbing instances of indirect aggression, social media platforms can create an environment that promotes healthy communication, encourages constructive interactions, and reduces the likelihood of users being exposed to harmful content. Enhancing the overall well-being of their users aligns with the platform's social responsibility and contributes to building a sustainable and loyal user community.

## Limitation and future work

This study has certain inevitable limitations. First, we must admit that we cannot employ the current state-of-the-art learning methods which are proposed every a few weeks, such as GPT, Llama (*Touvron et al., 2023*) and Gemma (*Banks & Warkentin, 2024*). Therefore,

the novel large pre-trained models are encouraged to be investigated in the field of cyber psychological and behavioral studies. Second, although deep learning model is employed for human behavior prediction, users' textual content is the main source to encode behavior. Next, the social media users' integrated behavior, namely basic characters, dynamic behavior and tweet content, can be a unified proxy to be feed into the behavioral encoder. This end-to-end behavior encoder will build a bridge between behavior prediction and deep learning. Third, this study uses specialized datasets specifically tailored to capture indirect aggression behaviors, which may not encompass the diversity found in standard datasets focused on direct aggression. Despite this, our methodology and findings remain valid and significant, as they provide novel insights into the less studied domain of indirect aggression.

### Funding

This study was supported by the National Natural Science Foundation of China under projects No. 62302319, the Humanity and Social Science Youth Foundation of Ministry of Education of China (Grant No. 22YJC860034), the R&D Program of Beijing Municipal Education Commission (Grant No. KM202210038002), and the Nankai University Asia Research Center Project (AS2310). The funders had no role in study design, data collection and analysis, decision to publish, or preparation of the manuscript.

### Grant Disclosures

The following grant information was disclosed by the authors:
National Natural Science Foundation of China: 62302319.
Humanity and Social Science Youth Foundation of Ministry of Education of China: 22YJC860034.
R&D Program of Beijing Municipal Education Commission: KM202210038002.
Nankai University Asia Research Center Project: AS2310.

### Competing Interests

The authors declare that they have no competing interests.

### Author Contributions

- Zhenkun Zhou conceived and designed the experiments, performed the experiments, analyzed the data, performed the computation work, prepared figures and/or tables, authored or reviewed drafts of the article, and approved the final draft.
- Mengli Yu conceived and designed the experiments, performed the experiments, analyzed the data, performed the computation work, prepared figures and/or tables, authored or reviewed drafts of the article, and approved the final draft.
- Xingyu Peng performed the experiments, analyzed the data, performed the computation work, prepared figures and/or tables, and approved the final draft.
- Yuxin He performed the experiments, analyzed the data, performed the computation work, prepared figures and/or tables, and approved the final draft.

## Ethics

The following information was supplied relating to ethical approvals (*i.e.*, approving body and any reference numbers):

School of Economics and Management of Beihang University granted Ethical approval to carry out the study within its facilities.

## Data Availability

The data and code are available at Zenodo: Zhenkun, Z. (2024). Dataset and code for predicting users's indirect aggression [Data set]. Zenodo. https://doi.org/10.5281/zenodo.10599993.

## Supplemental Information

Supplemental information for this article can be found online at http://dx.doi.org/10.7717/peerj-cs.2292#supplemental-information.

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
