# Peer review of "Predicting social media users’ indirect aggression through pre-trained models"

_PeerJ Computer Science, doi:10.7717/peerj-cs.2292_

## Round 0.1 · original submission · Major Revisions

Dear authors,

Thank you for submitting your article. Feedback from the reviewers is now available. It is not recommended that your article be published in its current format. However, we strongly recommend that you address the issues raised by the reviewers, especially those related to readability, experimental design and validity, and resubmit your paper after making the necessary changes.

Best wishes,

Reviewer 1 ·

Basic reporting

- The article should be revised for grammatical errors.

Experimental design

- How were users' features extracted? Especially dynamic features. What is meant by log(DAY S + 1), log(NT + 1), etc.?
- How were the features extracted from the day-level feature vector obtained? Uncertainty must be resolved.
-It should be explained how the K-encoder combines information on the data in the study.
- Were the results of RoBERTa and ELECTRA taken within the scope of this study? If so, the results should be included in Table 2 and Figure 5.
- Why was the training and test dataset chosen as 70:30?

Validity of the findings

Indirect aggression on social media is a new research topic. The differences in the method proposed in this paper from the limited studies on the subject and the innovation it will bring to the literature are not clearly stated. Contributions should be increased by adding the novelty of the data set and the innovative aspects of the method to the general statements in the theoretical contribution section.

Additional comments

In this study, a method based on ERNIE is proposed to solve the problem of indirect aggression in social media. The problem was classified into 3 different clusters by creating features considering the activities of Weibo users. The paper should be reviewed and necessary adjustments should be made, taking into account the evaluations I have made under different headings.

Cite this review as

Reviewer 2 ·

Basic reporting

This manuscript designs a model to predict social exclusion, malicious humour, and guilt induction online. The manuscript has a complete structure and the professional English is used. Experiment results could fully verify the effectiveness of the model. The research results are credible.

Experimental design

There is not comparative experiment of current deep learning-based solutions to this problem.

Validity of the findings

no comment.

Additional comments

no comment.

Annotated reviews are not available for download in order to protect the identity of reviewers who chose to remain anonymous.
Cite this review as

·

Basic reporting

Language is well written, but a thorough proofreading is recommended to fix minor grammatical errors. The following are the changes that are recommended to consider further

1. Provide a tabular representation of the existing works in the literature and differentiate how the proposed model is significant compared to other existing models for Predicting social media users' indirect aggression
2. Provide flow chart/pseudocode for the proposed approach

3. Comparison study/experiments need to be elaborated.

4. How the proposed model is efficient while utilizing resources, needs to be addressed.

5. Future works and conclusion needs to be changed.

Experimental design

Rigorous experimental analysis is required

Validity of the findings

Validity of the findings are worthy but needs to use diverse datasets

---

## Round 0.2 · accepted · Accept

Dear authors,

Thank you for the revision and for clearly addressing all the reviewers' comments. I confirm that the paper is improved. Your paper is now acceptable for publication in light of this revision.

Best wishes,

Reviewer 1 ·

Basic reporting

No comments

Experimental design

No comments

Validity of the findings

No comments

Additional comments

The authors have made the necessary corrections, taking into account the reviewer's evaluations.

Cite this review as

·

Basic reporting

Good

Experimental design

Good

Validity of the findings

Good